# $^{137}$Cesium ($^{137}$Cs) assessment in wild boars from northwestern Italy

**Daniele Pattono**[1]*, **Alessandro Mannelli**[1], **Alessandra Dalmasso**[1], **Riccardo Orusa**[2], **Massimo Faure Ragani**[3], **Maria Teresa Bottero**[1]

**1** Department of Veterinary Sciences, Largo Paolo Braccini 2 I-10095, Grugliasco (TO), Italy, **2** Istituto Zooprofilattico Sperimentale di Piemonte, Liguria e Valle d'Aosta, Aosta Section, Località Amerique 7/G, 11020 Quart (AO), Italy, **3** Agenzia Regionale per la Protezione dell'Ambiente (ARPA) Valle d' Aosta, Località La Maladière, 48, 11020 Saint Christophe (AO), Italy

* daniele.pattono@unito.it

## Abstract

Radionuclide contamination is a serious health issue caused by nuclear experiments and plant accidents, as seen for the Chernobyl and Fukushima nuclear plants. Italy has been especially interested in northwestern alpine regions, as have several other nations. The aim of this work was to indagate $^{134}$Cs and $^{137}$Cs contamination in wild boars, which were considered bioindicators sampled in the Chisone/Germanasca Valley and the Pellice Valley districts (Piedmont, Italy) in two hunting seasons (2014 and 2016). In the 2014 season, only the livers of the animals (n = 48) were sampled, whereas in 2016, five different anatomical sampling sites were sampled for each animal (n = 16). The analyses were conducted in an accredited laboratory (Agenzia Regionale per la Protezione dell'Ambiente–ARPA) by the aid of an HPGe detector (Ortec) with a relative efficiency of 50%. In general, the contamination levels registered in 2014 were under the detection limit for $^{134}$Cs and low for $^{137}$Cs (Chisone/Germanasca valley: min: 0.0, max: 23.9 median 11.0 Bq/kg vs Pellice valley: min 0, max: 31.7, median: 9.6 Bq/kg) and no health concern can be supposed. In the first-year samples, the liver showed a negative correlation between age and contamination level. In the second year of sampling, low levels were confirmed (min: 3.1 Bq/kg, max: 113.3; median 17.7 Bq/kg). Multiple sampling from the same animal showed that the diaphragm (median = 27.7 Bq/kg) kidney (27.4) and tongue (27.6) were more contaminated than the liver (17.7) and spleen (15.3). Moreover, a linear mixed model revealed a negative organ-by-age interaction, meaning that interorgan differences in contamination level were greater in younger (5–11 months) than in older (18–36 months) animals. Different feeding habits can be the explanation. Our paper shows that muscle sites (diaphragm and tongue) can be useful for radionuclide pollution surveillance in wild boar populations and that younger animals show more interorgan variability in contamination levels than older animals. More investigations are needed to confirm this correlation and to fulfill the request for more data to achieve better risk assessment.

**Data Availability Statement:** All relevant data are within the paper and its Supporting information files.

**Funding:** The author(s) received no specific funding for this work.

**Competing interests:** The authors have declared that no competing interests exist.

## Introduction

Radionuclide contamination as a threat to human health is receiving growing attention from the scientific community [1]. Sources of radionuclides may be natural, as in the case of radioactive minerals and sediments, or anthropogenic, such as the pollution derived from nuclear weapon experiments performed in the 1950s and 1960s and nuclear power plant accidents [2, 3]. Of the various anthropogenic radionuclides, the most frequently detected are 134Cesium (134Cs) ($T_{1/2}$ = 2.06 years) and 137Cesium (137Cs) ($T_{1/2}$ ranging from 30.17 to 69.3 years) [3–5]. Cs is generally found in the top few centimeters of the soil, with different vertical profiles for the different isotopes [6, 7]. In the soil environment, it can create bioavailable complexes and be absorbed by plants and mushrooms (secondary reservoirs) in the same way as potassium ions, thus becoming soil-independent sources of radionuclides and prolonging the contamination of sites [5, 8].

Anthropogenic sources of radionuclide contamination became the primary source in the second half of the twentieth century [2, 4, 6].

Considering the health consequences of anthropogenic radionuclide pollution, in 1957, the European Union (EU) set up regulations that were recently updated. The maximum level of 134Cs + 137Cs radioactivity in food products considered acceptable for consumption is 100 Bq/kg [6]. In addition, the new basic safety standard directives stated that environmental monitoring must be performed as an essential part of the management of emergency exposure and risk assessment, as in other fields [2, 9].

Moreover, assessing the effects of radionuclides on natural flora and fauna is important not only because of the potential impact of contamination on forest ecosystems but also because of their possible consumption by humans in the form of fruit, mushrooms, wild game meat and meat products [1, 4, 10]. In fact, it has been demonstrated that exposure to radionuclides through the consumption of contaminated food has the potential to affect a large part of the population and poses the most relevant radiological risk of chronic accumulation once measures have been taken [6, 7].

Within the biota of the forest ecosystem, wild boar (*Sus scrofa*) and earthworms are considered among the most reliable bioindicators for the study of these radionuclides [11–14]. Indeed, wild boars are considered hyperaccumulators of radionuclides [8]. There are many reasons for this; first, their diet is based on fungi and plants or insects in which radionuclides become bioavailable and that act as "secondary reservoirs". In addition, wild boars themselves can accumulate radionuclides, and their resorption rate is close to 100% [6, 15–17].

In the context of food safety, wild boars are regularly hunted for human consumption by the hunters themselves or sold on the national and international market. Moreover, these animals have the capability to travel long distances [13, 18–20] with a consequent propagation of risk; in fact, animals living in contaminated zones can be hunted far from the original zones. Furthermore, their meat can be sold in different areas. This fact means that hunter families and a wide number of consumers can be at risk of exposure to radionuclide contamination, and health problems must be considered a consequence of this environmental pollution. For these reasons, testing wild boars is essential to assess the potential for radionuclides to enter the food chain [1, 6, 10].

Muscle is the most common sampling site, but many authors have stressed that more data are needed on the distribution of radionuclides in the whole body to define safety limits more accurately, taking into account the distribution in the different anatomical systems and that some of the radionuclides are normally consumed [1, 6]. This is an important aspect if we consider that biodistribution and biological half-life depend on the organ [21].

In all Europe the attention on radionuclides contamination raised after two main nuclear plant accidents: Chernobyl and Fukushima nuclear plants accidents. Both nuclear plants accidents lead to a severe contamination across Europe of Cs isotope, $^{134}$Cesium ($^{134}$Cs) and $^{137}$Cesium ($^{137}$Cs) even if the contamination levels varied among regions [21]. With regard to Italy, considerable levels of $^{137}$Cs contamination were detected in the northwestern alpine region, in addition to natural sources of radionuclides. The region of Piedmont was affected; in particular, certain valleys, such as the Sesia Valley District [22, 23]. Some initial investigations to ascertain radionuclide contamination were conducted in Piedmont [22], but this was not followed up by detailed studies to assess radioactive contamination levels for the monitoring of contamination, and these data are still missing. In addition, evaluating the contamination of wild boar in these valleys provides us with valuable information for assessing the risk associated with the consumption of contaminated wild boar meat products, as well as data about radionuclide risk assessments requested by EU Regulations [9, 18, 24]. The output of this study will be useful for future sampling campaigns by helping to optimize the collection of data and to establish safe hunting management strategies [2, 13, 18].

The aim of this work was to determine the radioactive contamination of wild boar regularly hunted in the districts of the Pellice and Chisone/Germanasca Valleys in Piedmont, Northwest Italy (Fig 1), within the Western Alps. In the 2014 hunting season, we collected data on $^{134}$Cs and $^{137}$Cs contamination of the liver, whereas in the 2016 season, we collected data on the distribution of $^{137}$Cs in five different anatomical systems to study the distribution of radionuclide contamination in wild boar.

## Material and methods

All samples came from wild boars killed during two hunting seasons (2014 and 2016). The hunters were responsible for providing the anatomical samples collected at the slaughterhouse for the surveillance campaign carried out over these years.

In the first year (2014), liver samples (20–25 gr each sample) were collected in plastic bags by hunters from 48 wild boars (25 males and 23 females). The samples came from two valley districts in the Piedmont: the Chisone/Germanasca Valley (20 samples) and the Pellice Valley (28 samples). The ages were > 17 months (21 samples) and < 12 months (27 samples).

Two years later (2016), we obtained samples (10–100 gr depending on the anatomical site) from five different anatomical sites derived from 16 wild boars (6 males and 10 females) hunted in Pellice Valley District within the same hunting period. The sites comprised two muscular sites (diaphragm and tongue), kidney, spleen and liver. These sites of interest were chosen in light of the literature on radiocaesium contamination [1, 16] and to cover the main body systems, namely, the emunctory, digestive and hematopoietic systems.

$^{134}$Cs and $^{137}$Cs radioactivity was determined by gamma-spectrometric measurements performed by an accredited laboratory using an HPGe detector (Ortec) with a relative efficiency of 50%.

Tissue samples were stored at -20°C until the time of assay. Individual samples were dried at 105 °C to constant weight, then crushed in a cross-beater mill, and thoroughly mixed until homogeneous. Then the samples were placed in 20 cc to 100 cc polyethylene cups in contact with the detector End-Cap. The counting time was 60000 sec. The minimum detectable activity (MDA) was 1 Bq/kg (fresh weight) for both radionuclides. The calibration of the instrument/geometry was performed with a 100 cc certified multigamma source, and the Monte Carlo software EFFTRAN was used to evaluate the efficiency transfer due to the density and volume differences between the calibration source and the samples. All contributions to the

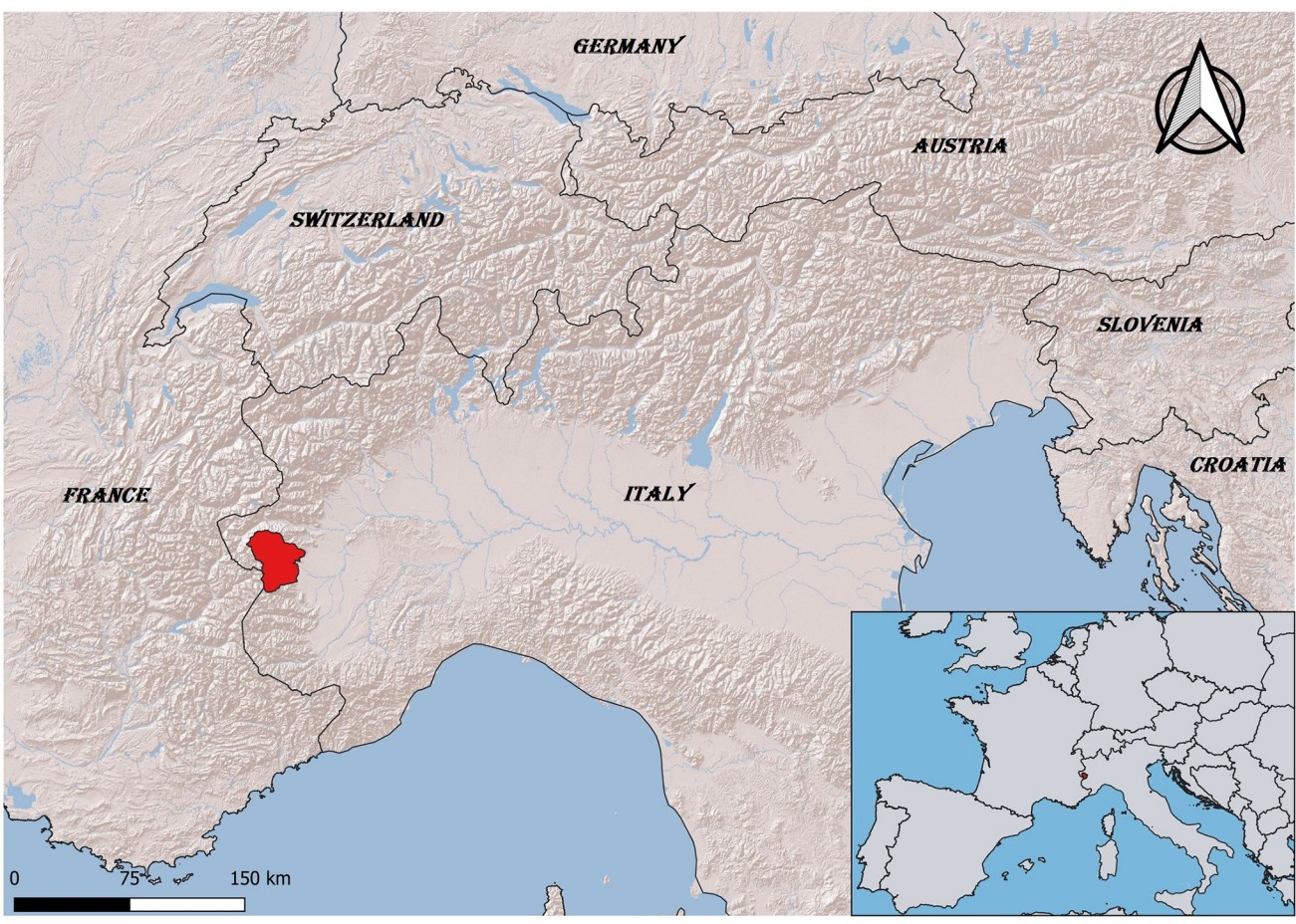

**Fig 1. Map of Chisone/Gremanasca and Pellice Valleys (Northwest Italy).**

uncertainty were considered: counting, gamma emission, efficiency, efficiency transfer, repeatability, and instrumental stability.

## Statistical analysis

The concentrations of $^{137}$Cs in wild boars' livers (2014) and in in five different anatomical sites (2016) were analyzed by the UNIVARIATE procedure in SAS System ver. 9.4 [25]. The Shapiro–Wilk test was used to check normality. The FREQ procedure was used to generate a distribution of the age of wild boars. Subsequently, individuals were divided into two age groups: 5 to 11, and 18 to 90 months of age. We used the MEANS procedure to calculate median, first and third quartiles (Q1, Q3) of the concentrations of $^{137}$Cs, since these statistics are not affected by the distribution of the data [25]. Furthermore, for descriptive purposes, we calculated minimum and maximum, mean, standard error and 95% confidence limits of the mean, after dividing the data by year, age group and sex of wild boars. For the second year of the study, descriptive statistics were obtained for each tested organ.

Wilcoxon two-sample tests were used to compare median concentrations from boars hunted in 2014, in the Chisone Germanasca Valley District with those hunted in the Pellice Valley Districts (PROC NPAR1WAY) and females vs. males. The correlation between the

concentrations of $^{137}$Cs in the liver and wild boar age was tested using the nonparametric correlation coefficient Kendall's tau (PROC CORR).

For data obtained in 2016, a linear mixed statistical model (PROC MIXED) was used to estimate the association between the tested organ, age group, and sex of wild boar and the natural log transformation of the concentrations of $^{137}$Cs (log$^{137}$Cs). The effects of these predictors were included in the model as fixed, as in a linear regression. We also considered that there might be variations in log$^{137}$Cs across wild boars. Furthermore, measurements, which we made on different organs belonging to the same 16 animals, were not independent. This needs to be taken into account in the statistical analysis. Since there was no interest in comparing specific individuals, between-wild boar variation was included in the model as a random effect [26]. Age group by tested organ interaction terms was included to test whether the association between organs and log$^{137}$Cs differed across age groups. Statistical significance was fixed at the 0.05 level.

## Results

As stated in the introduction, the sampling carried out in the 2014 hunting season was to obtain information on $^{134}$Cs and $^{137}$Cs levels in wild boars caught in two distinct mountain districts: the Chisone/Germanasca Valley and the Pellice Valley (Fig 1).

In 2014, $^{134}$Cs levels were always below the sensitivity threshold of the gamma-spectrometer, in agreement with a report from the national control authority in which levels of this contaminant in the soil were also below the sensitivity threshold for this method [27].

In the same year the median (first, third quartiles) level of $^{137}$Cs (Table 1 and S1 Table) in wild boar liver samples from the Chisone/Germanasca Valley was 11.05 Bq/kg (6.3, 15.7) vs. 9.6 (7.1, 17.8) for the Pellice Valley. This difference was not statistically significant (P = 0.98). Moreover, there was no significant difference between the liver $^{137}$Cs contamination levels in male vs. female boars hunted in 2014 (P = 0.46, Table 1). On the other hand, in 2014, we found a negative correlation between animal age and $^{137}$Cs liver contamination levels (Kendall's tau = -0.21, P < 0.05).

In the second year (2016), considering the results of the previous sampling, we focused only on $^{137}$Cs. Multiple anatomical sites were sampled to permit a comparison between three physiological systems, namely, the emunctory, digestive and hematopoietic systems. The results are summarized in Tables 2 and 3 and S2 Table. In general, the levels were very low for all the sampling sites.

The contamination with $^{137}$Cs (Fig 2) was greatest in organs from three wild boars of seven months of age (id = 5, 6, 7), with a maximum value of 279.1 Bg/kg in a diaphragm. Relatively high $^{137}$Cs levels were detected in only one animal of the older age group ($\geq$ 18 months; id = 10). Differences in $^{137}$Cs among organs were particularly wide in the three most contaminated, young wild boars. However, in relative terms, $^{137}$Cs in the diaphragm of

**Table 1. Radioactivity levels of $^{137}$Cesium ($^{137}$Cs, Bq/kg) in liver of 48 wild boars during the first sampling season (2014) in Chisone/Germanasca and Pellice Valleys, by age class and sex.** n = sample size; Q1: first quartile; Q3: third quartile; se: standard error; CL's: confidence limits of the mean.

| Age class, months (n) | Median value (Q1, Q3) | Minimum, maximum | Mean (se) | 95% CL's |
|---|---|---|---|---|
| 5, 11 (27) | 11.8 (7.7, 18.2) | 0.0, 39.7 | 13.3 (1.7) | 9.8, 16.8 |
| > 11 (21) | 9.0 (5.8, 14.2) | 0.0, 22.8 | 9.7 (1.4) | 6.8, 12.7 |
| **Sex (n)** | | | | |
| Female (23) | 9.1 (6.6, 14.2) | 0, 39.7 | 11.2 (1.7) | 7.7, 14.6 |
| Male (25) | 11.1 (6.9, 18.1) | 0, 31.7 | 12.3 (1.6) | 8.9, 15.6 |

**Table 2. Radioactivity levels of $^{137}$Cesium ($^{137}$Cs, Bq/kg) in 16 wild boars, in tongue, diaphragm, liver, spleen (results from all organs combined), during the second sampling season in Pellice Valley, 2016 by age class and sex.** n = number of observations; Q1: first quartile; Q3: third quartile; se: standard error; CL's: confidence limits of the mean.

| Age class, months (n) | Median value (Q1, Q3) | Minimum, maximum | Mean (se) | 95% CL's |
|---|---|---|---|---|
| 5, 11 (45) | 29.0 (18.9, 96.4) | 2.4, 279.1 | 62.1 (9.8) | 42.3, 81.8 |
| > 11 (35) | 15.0 (8.3, 27.2) | 3.1, 83.0 | 21.6 (3.3) | 14.9, 28.3 |
| **Sex (n)** | | | | |
| Female (50) | 27.9 (16.5, 61.5) | 2.4, 279.1 | 50.3 (8.1) | 33.9, 66.6 |
| Male (30) | 15.6 (8.3, 27.2) | 3.1, 199.6 | 34.5 (8.9) | 16.3, 52.7 |

**Table 3. Radioactivity levels of $^{137}$Cesium ($^{137}$Cs, Bq/kg) in different anatomical sites of 16 wild boars during the second sampling season in Pellice Valley, 2016.** Q1: first quartile; Q3: third quartile; se: standard error; CL's: confidence limits of the mean.

| Tested organ | Median value (Q1, Q3) | Minimum, maximum | Mean (se) | 95% CL's |
|---|---|---|---|---|
| Tongue | 27.4 (16.1, 40.7) | 3.4, 144.6 | 43.5 (11.5) | 19.0, 68.0 |
| Diaphragm | 27.7 (12.7, 52.9) | 4.2, 279.1 | 60.7 (20.0) | 18.2, 103.3 |
| Liver | 17.7 (10.3, 41.4) | 3.1. 113.3 | 32.9 (8.8) | 14.2, 51.6 |
| Spleen | 15.3 (8.2, 30.4) | 2.4, 109.1 | 29.4 (8.3) | 11.7, 47.1 |
| Kidney | 27.6 (17.1, 50.0) | 4.5, 192.0 | 55.2 (16.0) | 21.0, 89.4 |

another young wild boar (id = 2) was 5.6 times greater than in the spleen of the same animal (46.1 vas 8.2 Bq/kg).

The Shapiro–Wilk test indicated a departure from normality (W = 0.71) of the distribution of $^{137}$CS in 2016, which justified the natural log transformation (log$^{137}$Cs) to fit a normal distribution (W = 0.98) to be used in the linear mixed model. The spleen, which was characterized by the lowest contamination levels (Table 3), was used as the reference organ in the analysis. The final model successfully converged, and the results, which are shown in Table 4, indicated that differences between log$^{137}$Cs across organs were greatest in younger wild boars. This was demonstrated by significant, negative diaphragm-by-age, and kidney-by-age interaction terms. There was also a negative tongue-by-age interaction, although it was not significant at

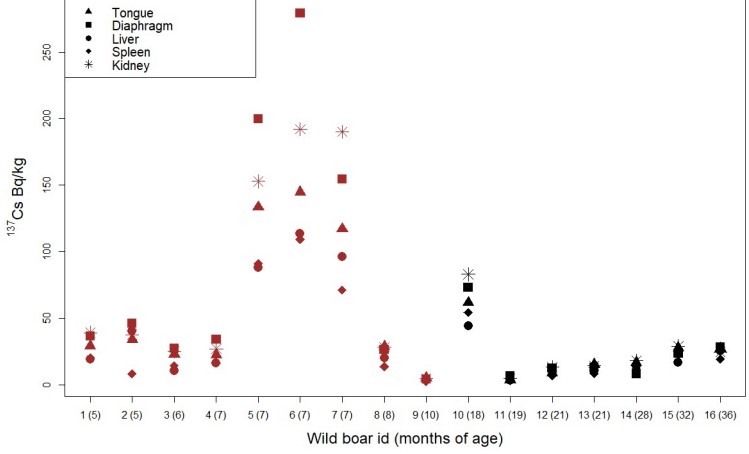

**Fig 2. Concentration of $^{137}$Cesium ($^{137}$Cs) in different organs from wild boars in northern Italy in 2016.** Black symbols: wild boars aged > 18 months; brown symbols: wild boars aged < 11 months.

**Table 4. Results of a mixed linear statistical model of the association between tested organ, age group, and the natural log of 137Cs (log137Cs), in wild boars in Italy, in 2016.** The spleen was used as the reference organ. Individuals were divided into two age groups: > 18 months and < 11 months as the reference group. Consequently, the negative interaction sign indicates that the increased log137Cs levels in certain organs, in comparison with the spleen, were greatest in young wild boars.

| Fixed effect | Estimate | Standard Error | P |
|---|---|---|---|
| Intercept | 2.87 | 0.60 | 0.00 |
| Diaphragm | 0.82 | 0.10 | < .0001 |
| Liver | 0.25 | 0.10 | 0.02 |
| Tongue | 0.58 | 0.10 | < .0001 |
| Kidney | 0.74 | 0.10 | < .0001 |
| Spleen | 0 | . | . |
| Female | 0.23 | 0.61 | 0.70 |
| Male | 0 | . | . |
| Age | -0.46 | 0.60 | 0.44 |
| Diaphragm * Age | -0.51 | 0.15 | 0.002 |
| Liver * Age | -0.21 | 0.15 | 0.19 |
| Tongue * Age | -0.29 | 0.15 | 0.07 |
| Kidney * Age | -0.30 | 0.15 | 0.05 |
| Spleen * Age | 0 | . | . |

the 0.05 level. There was no significant association between wild boar sex and log137Cs (Table 4). There was no significant interaction between sex and organ or between sex and wild boar age; therefore, the corresponding terms were not retained in the final model.

## Discussion

The aim of this paper was to determinate Cs contamination in two Valleys of Piedmont region several years after Chernobyl and Fukushima nuclear plants accidents. At the time of the Chernobyl accident, the Agenzia Regionale per la Protezione dell'Ambiente of the Piedmont region (ARPA), a public agency for environmental protection, evaluated a radioactive fallout of 137Cs ranging from 31500 (north alpine valleys of the Piedmont) to values lower than 4000 Bq/m² in the southern valley. In the investigated valleys, values were between 4000 and 7800 Bq/m² [27]. Even if in food of animal origin, relatively low values were registered (farm Milk in 1990 value approximately 1.2 Bq/l and values lower than 0.2 Bq/l in 2011) in particular situations (milk produced from highland pastures), values of approximately 20–30 Bq/l were still found at the time of the investigation [27].

The results concerning 134Cs levels show in our opinion that the nuclear power accidents did not result in the contamination of these two valleys. These findings are in agreement with a report from the national control authority in which levels of this contaminant in the soil were also below the sensitivity threshold for this method [27].

The contamination levels of 137Cs in 2014 are similar to values reported for 137Cs in wild boar meat from Piedmont in 2012 (approximately 10 Bg/kg) [27] but very different from findings for other European countries in the same period. For example, the mean contamination level reported for wild boar hunted in Switzerland was 1084 Bq/kg, and the maximum was 2420 Bq/kg [16], while another study reported mean values eightfold higher for central Europe, in the range of 20 kBq/kg [8]. In Bavaria wild boar, meats exceeded the regulatory limit by 1–2 orders of magnitude. High contamination values (min 0.37 kBq/kg max 14 kBq/kg–median 1.7 kBq/kg) were still registered several years later (fall 2019 –spring 2020) in the tongue [28].

Thus, the range of values for animals across Europe is extremely wide [5, 8]. The contamination was widely spread across Europe and in agreement with other sites showed a slower decline than what could be expected considering the half-life of $^{137}$Cs in wild boar meat. This phenomenon is called the "wild boar paradox" [28]. In addition, it was demonstrated also in a retrospective study for vegetables, water and soils [29]. Recently, Saito et al. [30] explained that the mechanical action of wild boar rooting behavior is responsible for longer persistence. In fact, this peculiar feeding behavior is responsible for changing the depth of $^{137}$Cs and chemical or biological interactions with the organic layer and mineral soil components. The consequences are changes in bioavailability and vertical and horizontal/surface distributions in contaminated areas.

In relation to the consumption of wild boar meat, considering the statutory limit of 600 Bq/kg for $^{137}$Cs [7], the low contamination levels detected in samples collected in 2014 confirm the lack of any health concerns related to wild boar hunted in the two valleys studied. Moreover, the risks associated with eating this meat are reduced considering that levels can be lower because the boars were hunted in the autumn when levels are higher [13, 17, 20]. This means that in other periods, the contamination can be lower. Another recommendation to decrease the rate of consumption and consequently the health-related risk is to eat cooked meat instead of raw meat preparations (salami or raw meat dishes) because the cooking process appears to decrease the effects of radionuclide contamination in food [11].

Other considerations can be done comparing different classes, *e.g.* male vs. female, different sampling sites or different ages. Comparing the liver $^{137}$Cs contamination levels in male vs. female boars hunted in 2014, no significant differences were found (P = 0.46). While this confirms the finding of some authors [15], it does not agree with those of others [31]; however, this result remains controversial because few data are available in the literature about comparisons between male and female subjects. The same consideration must be made in relation to the provenience of the animals (in 2014).

A negative correlation between animal age and $^{137}$Cs liver contamination levels, in 2014, indicated that samples from younger animals tend to be more contaminated than those from older animals, as described for other species [19]. Differences in feeding habits in quality and quantity (younger animals consume higher quantities of contaminated feed) may explain this finding [7, 13, 18, 29, 32]. If this is so, a new approach to risk assessment should be devised, with younger animals being considered the preferred target for the sampling campaigns.

In general, the diaphragm, kidney, and tongue were the most heavily contaminated organs, whereas the spleen and liver were characterized by lower values (Table 3; Fig 2). The interorgan differences in contamination levels were most evident in young animals. There is no literature available with which to compare these findings, and more studies are needed.

If we consider the distribution in the literature, a similar distribution was described by other authors for these organs. Gulakov [16] found that muscles and kidneys were more contaminated than the liver and spleen, but no information was provided about animal age. In another study looking at bovine samples, mean values varied greatly between muscle sites (*e.g.*, contamination levels in the tongue were twice those of the diaphragm), with the highest found in the liver and the lowest in the kidney [33]. In Bavarian wild boar samples, the contamination of tongue and other unspecified muscles was higher than our findings (min 0.37 kBq/kg, max 14 kBq/kg–median 1.7 kBq/kg) [28]. From our point of view, the different contamination levels among anatomical sites found can be ascribed to a variety of factors [34–36]. The first one can be the different equilibrium times for $^{137}$Cs. The higher contamination of muscles can also be explained considering the different chemical composition of this specific anatomical site. It is a tissue with high Potassium (K) content and Cs shows similar biodynamic patterns

being in the same group of the periodic table (Group 1 alkali metals). This is in accordance with other papers [37, 38].

Explanations for the different contamination levels among the papers can be differences between monogastric vs. poligastric animals, differences in feeding habits, differences in the levels of feed contamination, and the heterogeneous deposition of radionuclide fallout within contaminated areas [7, 13, 17, 18, 27]. The results also indicate that the muscles sites are associated with the highest level of contamination considering edible tissues [1].

Also, the kidney seems to show higher contamination. This issue is interesting because distribution has been studied in several animals and insects but not in wild boar [21, 34, 36].

Several final considerations can be summarized from the discussion. New insight into contamination levels, food safety and distribution among organs can be used for the risk assessment of radionuclide contamination in wild boar. First, in these valleys, contamination levels seem to be low, which confirms the lower fallout of Cs in the southern parts of the Piedmont [27]. Moreover, it has been proven that muscular site sampling can be considered a reliable site for checking contamination in the case of ordinary and extraordinary surveillance. The advantage is that muscles are already considered a routine sampling site for hunters for trichinellosis surveillance programs, and for this reason, there is no need for training [39].

The comparison among different organs confirms that the kidney can also be useful [1, 13, 18]. Unfortunately, few other studies are available in the present literature for other organs in field surveillance with which to compare our results, a problem also highlighted by Saito et al. [17].

Finally, our data show that organs from certain younger animals were the most contaminated, so we can suggest that younger animals are more useful for risk assessment. This can be useful for surveillance campaigns.

## Conclusions

Radionuclides are gaining attention for their importance and potential consequences for public health. The pollution resulting from nuclear power plant accidents (e.g., Chernobyl) is still present and continues to cause a small increased risk of stochastic radiation effects (cancer and heritable effects) [7]. To provide a high level of protection from this risk, mandatory risk assessments must be performed using reliable data [4, 18] and in a regular manner due to the very long half-lives of certain radionuclides, such as $^{137}$Cs [8].

Wild boars are considered to be valuable bioindicators and a significant source of contamination for people consuming their meat products [7, 10]. At the present time, the request for data is only partially fulfilled [1].

Our research shows that in these valleys, the situation is good with low contamination levels. An important finding of our research indicates that juvenile animals have slightly higher contamination and that variations between sampling points are more detectable in these individuals. In accordance with recent legislation [9], more work is needed to extend monitoring activities to other radionuclides to a greater number of animals, and different matrices such soil, food material and feces are needed to study the phenomenon more thoroughly to confirm the statistically significant correlations found here. The analysis should also be extended to cover additional edible animal parts and other species for a more complete evaluation of game meat safety for human consumption, thus raising the level of food safety associated with game meat and game meat products [16, 33]. In addition, hunters might have available safer and more reliable instructions given by authorities for hunting campaigns (*e.g.*, young animals can be more useful for assessment and surveillance). At present, the information gathered about $^{137}$Cs contamination helps consumers and producers to a safer consumption of this kind of

meat by cooking it or using it for cooked products instead of raw fermented products such as salami, raw ham or tartare or suggests the exclusion of some edible organs.

## Supporting information

**S1 Table. First-year sampling data.**
(XLS)

**S2 Table. Second-year sampling data.**
(XLS)

## Acknowledgments

We would like to thank Mr. Marco Giovo and Dr. Elisa Dalmas for their help during the sampling phase.

## Author Contributions

**Data curation:** Alessandro Mannelli.

**Formal analysis:** Massimo Faure Ragani.

**Project administration:** Daniele Pattono, Alessandra Dalmasso, Riccardo Orusa, Maria Teresa Bottero.

**Writing – original draft:** Daniele Pattono, Alessandro Mannelli.

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
