## [Decision Letter · Decision Letter 0]

20 Oct 2023

PONE-D-23-24864Radionuclide assessment in wild boars from Northwest Italy.PLOS ONE

Dear Dr. pattono,

Thank you for submitting your manuscript to PLOS ONE. After careful consideration, we feel that it has merit but does not fully meet PLOS ONE’s publication criteria as it currently stands. Therefore, we invite you to submit a revised version of the manuscript that addresses the points raised during the review process.

We look forward to receiving your revised manuscript.

Kind regards,

Mohamad Syazwan Mohd Sanusi

Academic Editor

PLOS ONE

Reviewers' comments:

Reviewer's Responses to Questions

**Comments to the Author**

1. Is the manuscript technically sound, and do the data support the conclusions?

Reviewer #1: Partly

Reviewer #2: Yes

Reviewer #3: Yes

Reviewer #4: Partly

Reviewer #5: Yes

2. Has the statistical analysis been performed appropriately and rigorously? 

Reviewer #1: N/A

Reviewer #2: Yes

Reviewer #3: I Don't Know

Reviewer #4: I Don't Know

Reviewer #5: Yes

3. Have the authors made all data underlying the findings in their manuscript fully available?

Reviewer #1: Yes

Reviewer #2: Yes

Reviewer #3: Yes

Reviewer #4: No

Reviewer #5: Yes

4. Is the manuscript presented in an intelligible fashion and written in standard English?

Reviewer #1: Yes

Reviewer #2: Yes

Reviewer #3: Yes

Reviewer #4: Yes

Reviewer #5: Yes

5. Review Comments to the Author

Reviewer #1: Although the work is interesting, the manuscript requires serious modifications in several places before it can be recommended for publication in PLOS ONE.

• More investigations are needed to confirm the results and to fulfill the request for more data in order to achieve better 59 risk assessment.

• Language and sentence construction are another areas which need thorough improvement.

• The references cover the the past activities in the field. Particularly, results and discussion should be re-written in more details using current literatures.

• Figure 1 should be drawn as the graph linearly.

Reviewer #2: The manuscript titled "Radionuclide Assessment in Wild Boars from Northwest Italy" addresses the issue of radionuclide contamination in wild boars and its potential impact on human health. The study is conducted in the northwest regions of Italy, which have been affected by nuclear events, including the Chernobyl accident. Radionuclides, such as cesium-137 (137Cs), were detected in samples of wild boars hunted in the studied regions. The study highlights that radionuclide contamination can pose a threat to human health, whether from natural sources or anthropogenic sources like nuclear accidents. European Union regulations establish safe limits for radioactivity in food products, including wild boar meat, to ensure food safety. The manuscript examines the concentrations of 137Cs in the livers of wild boars collected from different regions and hunting seasons. The results reveal that, despite the detected contamination, the levels are below regulatory limits and therefore do not pose a significant concern for human health, especially considering the cooking of wild boar meat, which can reduce the effects of radionuclide contamination. Furthermore, the study emphasizes the importance of considering the age of the animals in contamination assessment, noting a negative correlation between the age of the animals and contamination levels.

This article provides a valuable contribution to the understanding of radionuclide contamination in wild boars in the northwest region of Italy. The results suggest that the contamination levels found are below established regulatory limits, which is reassuring from a food safety perspective. Additionally, the analysis of the correlation between age and contamination is an interesting and relevant finding for research. However, it is important to note that the sampling is limited in terms of sample size and the variety of organs analyzed. To obtain a more comprehensive and representative picture, further studies with larger samples and a comprehensive analysis of different organs would be beneficial. In summary, this article offers valuable insights into radionuclide contamination in wild boars, emphasizing the importance of food safety and the need to continue monitoring and assessing the risks associated with radioactive contamination in food.

After the review process that has already taken place, I am satisfied with the manuscript as it stands. My comments and suggestions have been addressed by the other reviewers, which has enriched the overall quality of the article. However, there are still some minor details that I would like to suggest the authors consider adding to make the study more comprehensive.

1) Firstly, it would be beneficial if the authors could provide a discussion of the practical implications of the findings (just one sentence). For example, they could discuss how the findings of this study may influence food safety policies related to wild boar meat in the northwest region of Italy and elsewhere. This would help readers better understand the real-world impact of these findings/importance of the study.

2) Furthermore, it would be helpful if the authors could briefly mention any limitations of the study, such as sampling constraints or uncertainties associated with radioactivity measurement methods (Methods section). Acknowledging these limitations would provide a more complete and honest view of the work.

3) In addiction, given the relevance of the results to food safety, the authors may want to highlight some practical guidelines or recommendations for hunters, regulators, or consumers based on the study's findings.

4) The sentence "In our opinion" appears three times in the text. One of those would be better replaced with "In our point of view". That would make the manuscript more clearear.

5)Lately, I suggest inserting an image/map of Italy and the two valleys of the sampling place (Chisone/Germanasca and the Pellice Valley). This would be great to localize the readers (Line 159-161).

These additional suggestions would further strengthen the article and make it more informative and applicable to the scientific community and those interested in food safety and the management of radionuclide contamination in game products.

Reviewer #3: This paper presents a critical evaluation of the article titled "Radionuclide assessment in wild boars from Northwest Italy". The article under scrutiny addresses a significant issue and offers valuable insights. Upon careful examination, it is determined that the overall assessment of this manuscript is accept, with only minor revisions required.

The abstract is excessively lengthy, comprising nearly 400 words and consisting of 24 lines in font size 11. It is advisable to condense and enhance its focus and precision on the study's outcomes.

The statistical analysis is perplexing, as I am unacquainted with the referenced statistical models. It would be advantageous to provide a concise explanation of the methodologies employed, in order to furnish the readers with insights into understanding the statistical analysis.

The presentation of the results lacks precision and should be more comprehensive. Additionally, if the authors opt to integrate the results with the discussion, it is imperative that they thoroughly analyze each result subsequent to its reporting.

The conclusion fails to effectively summarize the findings; rather, it appears to serve as a set of recommendations. Consequently, it is recommended that the conclusion be revised to recommendations. which are highly valuable and significant for future research endeavors.

Finally, the manuscript needs to be edited by a native English speaker to enhance its clarity and fluency.

Reviewer #4: The data presented in the article is interesting to read. However, I have some additional remarks and comments that are necessary:

1) Showing statistical analyses is important, but it is necessary to see the initial data as well. Please provide supplementary activity concentrations of Cs-137 in organs for both seasons. Additionally, please specify whether the data is for dry or fresh weight.

2) The paper would be more complete if data for activity concentrations in fecal samples were available. This would provide some idea of the activity concentration range in organs.

3) It would be interesting to see any data describing soil contamination or radioactive fallout in Northwest Italy.

4) High activity concentration can be simply described by considering different equilibrium times for Cs-137 in different organs. You can refer to any classical book for more information on this topic.

I hope you find these suggestions helpful for improving your article.

Best regards,

Reviewer #5: Dear Authors

Based on the replies to previous comments, the quality of the article has improved

The article contains interesting data for the scientific community.

however, the title of the article is misleading as only Cs-137 ( Cs-134) was measured in the publication and not the other radionuclides ( and the concentrations of the other radionuclides were not assessed) .

I suggest a clarification of the title of the article.

6. PLOS authors have the option to publish the peer review history of their article (what does this mean?). If published, this will include your full peer review and any attached files.

Reviewer #1: No

Reviewer #2: **Yes: **Sérgio José Gonçalves Jr.

Reviewer #3: No

Reviewer #4: No

Reviewer #5: No

---

## [Author Response · Author response to Decision Letter 0]

12 Dec 2023

Dear Editor and Reviewers,

thank you so much for the time you have dedicated to our manuscript. We agree that in the previous version of the manuscript, there were important concerns to resolve, and we truly thank you and the editorial office for the opportunity to answer to all the concerns raised by you and by the referees.

The answers are written in red, whereas the changes in the manuscript are highlighted in yellow Enclosed here, you find all the changes and the additions we made to answer the questions raised for the revision process.

We have revised our manuscript for English quality by a native English speaker. We attached here the certificate of the language revision.

Reviewer #1: Although the work is interesting, the manuscript requires serious modifications in several places before it can be recommended for publication in PLOS ONE.

• More investigations are needed to confirm the results and to fulfill the request for more data in order to achieve better 59 risk assessment.

Answer:

However, more sampling is needed for a complete risk assessment. Our data can be an objective starting point for a new sampling campaign. Data were not available for these valleys, especially liver samples, as stated by Ferri et al. (2017). In our opinion, our paper adds useful information to the current knowledge to improve more specific designs for risk assessment and therefore achieve better surveillance campaigns. Our intention is to continue to test for radionuclide pollution in wild species considering the important suggestion given by the reviewer

Language and sentence construction are other areas which need thorough improvement.

Answer:

The paper has been reviewed by a native English speaker, the certificate is attached.

From the revision agency “Your document was edited for correct English language, grammar, punctuation, and phrasing. In addition, we have made some changes to ensure consistency throughout the document. These changes are based on our extensive knowledge of what journals typically require.”

• The references cover the past activities in the field. Particularly, results and discussion should be re-written in more details using current literatures.

Answer: 

New paragraphs and new considerations have been added to the sections (Page 6 from line 126 to line 127, from Page 11 line 251 to Page 12 line 272, Page 14 from line 311 to line 313, Page 16 from line 351 to line 357, Page 16 from line 364 to line 365, Page 16 from line 366 to Page 17 line 384).

Stäger F, Zok D, Schiller AK, Feng B, Steinhauser G (2023). Disproportionately high contribution of 60 years old weapons-137Cs explain the persistence of radioactive contamination in Bavarian wild boars. Environ Sci technol 57, 13601-13611 http://doi.org/10121/acs.est3c03565

Endo S, Matsunami Y, Kajimoto T, Tanaka K, Suzuki M (2020). Internal exposure rate conversion coefficients and absorbed fraciont of mouse for 137Cs, 134Cs and 90Sr contamination in body. J Rad Res, 61 (4), 535-545. Doi:10.1093/jrr/rraa030

Mamyrbayeva AS, Baigazinov ZA, Lukashenko SN, Panitskiy AV, Karatayev SS, Shatrov AN, et al (2020). The trasfert of 241Am and 137Cs to the tissues of broilers’organs. PloS ONE, 15(7):e0235109. https://doi.org/10.1371/journal.pone.0235109

Baigazinov Z, Lukashenko S, Silybayeva B, Zharykbasova K, Bukabayeva Z, Muhamediarov N, et al (2023). The transfer of 137Cs and heavy metals to tissues within the organs of snails. Nature 13:15690 https://doi.org/10.1038/s41598-023-42580-6

Eissa F and E-Dein A (2023). Irradiated and radioactively contaminated foods: Analysis of EU RASFF notifications from 10097 to 2022. J Environ Radioact 270, 107315 https://doi.org/10.1016/j.envrad.2023.107315

Saito R, Wakiyama Y, Bontrager H, Nanba K, Beasley JC (2023). Alteration of the Caesium-137 soil profile by wild boar rooting after the Fukushima Daiichi Nuclear Power Plant accident. Environmental Challenges 12, 100728 http://doi.org/10.1016/j.envc.2023.100728

Isaksson M, Tondel M, Wålinder R, Rääf C (2021). Absorbed dose rate coefficients for 134 Cs and 137Cs with steady-state distribution in the human body: S-coefficients revisited. J Rad Prot 41, 1213-1227 https://doi.org/10.1088/1361-6498/ac2ec4

• Figure 1 should be drawn as the graph linearly.

Answer:

Figure n. 1 has been revised, and a more linear order has been signed for the samples.

Reviewer #2: The manuscript titled "Radionuclide Assessment in Wild Boars from Northwest Italy" addresses the issue of radionuclide contamination in wild boars and its potential impact on human health. The study is conducted in the northwest regions of Italy, which have been affected by nuclear events, including the Chernobyl accident. Radionuclides, such as cesium-137 (137Cs), were detected in samples of wild boars hunted in the studied regions. The study highlights that radionuclide contamination can pose a threat to human health, whether from natural sources or anthropogenic sources like nuclear accidents. European Union regulations establish safe limits for radioactivity in food products, including wild boar meat, to ensure food safety. The manuscript examines the concentrations of 137Cs in the livers of wild boars collected from different regions and hunting seasons. The results reveal that, despite the detected contamination, the levels are below regulatory limits and therefore do not pose a significant concern for human health, especially considering the cooking of wild boar meat, which can reduce the effects of radionuclide contamination. Furthermore, the study emphasizes the importance of considering the age of the animals in contamination assessment, noting a negative correlation between the age of the animals and contamination levels.

This article provides a valuable contribution to the understanding of radionuclide contamination in wild boars in the northwest region of Italy. The results suggest that the contamination levels found are below established regulatory limits, which is reassuring from a food safety perspective. Additionally, the analysis of the correlation between age and contamination is an interesting and relevant finding for research. However, it is important to note that the sampling is limited in terms of sample size and the variety of organs analyzed. To obtain a more comprehensive and representative picture, further studies with larger samples and a comprehensive analysis of different organs would be beneficial. In summary, this article offers valuable insights into radionuclide contamination in wild boars, emphasizing the importance of food safety and the need to continue monitoring and assessing the risks associated with radioactive contamination in food.

After the review process that has already taken place, I am satisfied with the manuscript as it stands. My comments and suggestions have been addressed by the other reviewers, which has enriched the overall quality of the article. However, there are still some minor details that I would like to suggest the authors consider adding to make the study more comprehensive.

1) Firstly, it would be beneficial if the authors could provide a discussion of the practical implications of the findings (just one sentence). For example, they could discuss how the findings of this study may influence food safety policies related to wild boar meat in the northwest region of Italy and elsewhere. This would help readers better understand the real-world impact of these findings/importance of the study.

Answer:

Requested suggestions have been added to the text (Page 6 from line 115 to line 118 and Page 18 from line 400 to line 404).

This fact means that hunter families and a wide number of consumers can be at risk of exposure to radionuclide contamination, and health problems must be considered a consequence of this environmental pollution.

Our research shows that in these valleys, the situation is good with low contamination levels. An important finding of our research indicates that juvenile animals have slightly higher contamination and that variations between sampling points are more detectable in these individuals.

2) Furthermore, it would be helpful if the authors could briefly mention any limitations of the study, such as sampling constraints or uncertainties associated with radioactivity measurement methods (Methods section). Acknowledging these limitations would provide a more complete and honest view of the work.

Answer:

We are aware of the limitation of the number of samples obtained and of the number of analyses conducted. The results support new sampling with a more specific design considering wild boars and other local wild species and can be useful for analyzing liver contamination (Ferri et al., 2017). The requested suggestions have been added to the text and more specific indications have been added to the methods section about measurements as requested by the reviewer (Page 8 from line 173 to line 175 and Page 8 from line 180 to Page 9 line 187, Page 17 from line 384 to line 385, Page 18 from line 406 to line 407).

134Cs and 137Cs radioactivity was determined by gamma-spectrometric measurements performed by an accredited laboratory using an HPGe detector (Ortec) with a relative efficiency of 50%.

The calibration of the instrument/geometry was performed with a 100 cc certified multigamma source, and the Monte Carlo software EFFTRAN was used to evaluate the efficiency transfer due to the density and volume differences between the calibration source and the samples. All contributions to the uncertainty were considered: counting, gamma emission, efficiency, efficiency transfer, repeatability, and instrumental stability.

different matrices such soil, food material and feces are needed

3) In addiction, given the relevance of the results to food safety, the authors may want to highlight some practical guidelines or recommendations for hunters, regulators, or consumers based on the study's findings.

Answer:

Requested suggestions have been added to the text (Page 13 from line 281 to line 285, Page 17 from line 382 to line 385, Page 18 from line 400 to line 404 and Page 18 from line 413 to Page 19 line 420).

the rate of consumption and consequently the health-related risk is to eat cooked meat instead of raw meat preparations (salami or raw meat dishes) because meat is usually consumed cooked and the cooking process appears to decrease the effects of radionuclide contamination in food [11].

Finally, our data show that organs from certain younger animals were the most contaminated, so we can suggest that younger animals are more useful for risk assessment. This can be useful for surveillance campaigns. 

Our research shows that in these valleys, the situation is good with low contamination levels. An important finding of our research indicates that juvenile animals have slightly higher contamination and that variations between sampling points are more detectable in these individuals.

The information gathered about Cs 137 contamination can help consumers and producers to a safer consumption of this kind of meat (e.g., cooking it or using it for cooked products instead of raw fermented products such as salami, raw ham or tartare or exclusion of edible organs). Moreover, hunters might have available safer and more reliable instructions given by authorities for hunting campaigns (e.g., young animals can be more useful for assessment and surveillance).

4) The sentence "In our opinion" appears three times in the text. One of those would be better replaced with "In our point of view". That would make the manuscript clearer.

Answer:

The sentence has been rewritten as requested (Page 16 line 354).

From our point of view, this can

5)Lately, I suggest inserting an image/map of Italy and the two valleys of the sampling place (Chisone/Germanasca and the Pellice Valley). This would be great to localize the readers (Line 159-161).

Answer:

A map of Valleys and north Italy has been attached in the paper (Fig. n. 1 Page 7 line 148 and a new numeration for Fig. n. 2 at Page 14 line 315).

These additional suggestions would further strengthen the article and make it more informative and applicable to the scientific community and those interested in food safety and the management of radionuclide contamination in game products.

Reviewer #3: This paper presents a critical evaluation of the article titled "Radionuclide assessment in wild boars from Northwest Italy". The article under scrutiny addresses a significant issue and offers valuable insights. Upon careful examination, it is determined that the overall assessment of this manuscript is accept, with only minor revisions required.

The abstract is excessively lengthy, comprising nearly 400 words and consisting of 24 lines in font size 11. It is advisable to condense and enhance its focus and precision on the study's outcomes.

Answer:

The abstract has been rewritten in a shorter way (from Page 2 line 23 to Page 3 line 50).

The statistical analysis is perplexing, as I am unacquainted with the referenced statistical models. It would be advantageous to provide a concise explanation of the methodologies employed, in order to furnish the readers with insights into understanding the statistical analysis.

Answer:

An explanation has been added to help readers better comprehend the statistical methods used (Page 10 line 209 to line 222). A new reference has been added at Page 23 from line 526 to line 528.

A linear mixed statistical model [25] was used to estimate the association between the tested organ, age group, and sex of wild boar and logCs. The effects of these predictors were included in the model as fixed, as in a linear regression. We also considered that there might be variations in logCs across wild boars. Furthermore, measurements, which we made on different organs belonging to the same 16 animals, were not independent. This needs to be taken into account in the statistical analysis. Since there was no interest in comparing specific individuals, between-wild boar variation was included in the model as a random effect [26]. Age group by tested organ interaction terms was included to test whether the association between organs and logCs differed across age groups. Statistical significance was fixed at the 0.05 level.

Dohoo, Ian Robert, S. Wayne Martin, and Henrik Stryhn. 2012. Methods In Epidemiologic Research. Charlottetown, P.E.I.: VER, Inc.

The presentation of the results lacks precision and should be more comprehensive. Additionally, if the authors opt to integrate the results with the discussion, it is imperative that they thoroughly analyze each result subsequent to its reporting.

Answer:

More parts have been added in order to be more precise and comprehensive (Page 11 line 245, Page 11 from line 251 to Page 12 line 272, Page 13 from line 281 to line 287, Page 13 from line 295 to line 298, Page 14 from line 304 to line 305, Page 14 from line 313 to Page 15 line 343).

The conclusion fails to effectively summarize the findings; rather, it appears to serve as a set of recommendations. Consequently, it is recommended that the conclusion be revised to recommendations. which are highly valuable and significant for future research endeavors

Answer:

The conclusions have been rewritten following the suggestions of the reviewer. We added paragraphs to summarize the results of the present paper. In the conclusions Page 18 from line 400 to line 404.

Our research shows that in these valleys, the situation is good with low contamination levels. An important finding of our research indicates that juvenile animals have slightly higher contamination and that variations between sampling points are more detectable in these individuals.

Finally, the manuscript needs to be edited by a native English speaker to enhance its clarity and fluency.

Answer:

The paper has been reviewed by a native English speaker, the certificate is attached.

From the revision agency “Your document was edited for correct English language, grammar, punctuation, and phrasing. In addition, we have made some changes to ensure consistency throughout the document. These changes are based on our extensive knowledge of what journals typically require.”

Reviewer #4: The data presen

---

## [Decision Letter · Decision Letter 1]

6 Mar 2024

PONE-D-23-24864R1137Cesium (137Cs) assessment in wild boars from northwestern ItalyPLOS ONE

Dear Dr. pattono,

Thank you for submitting your manuscript to PLOS ONE. After careful consideration, we feel that it has merit but does not fully meet PLOS ONE’s publication criteria as it currently stands. Therefore, we invite you to submit a revised version of the manuscript that addresses the points raised during the review process.

We look forward to receiving your revised manuscript.

Kind regards,

Mohamad Syazwan Mohd Sanusi

Academic Editor

PLOS ONE

Journal Requirements:

Additional Editor Comments:

Decision: Minor Revision

General Comments: The title of the work, "137Cesium (137Cs) assessment in wild boars from northwestern Italy," is worthy of study. The proposed method, with large samplings and targeted organ samples of the wild boars (48 for liver and 16 samples for each of the 5 types of samples), is technically sound. The method is sufficient, and the qualitative analysis approach investigating the correlation between Cs levels and age, and different types of tissue organs, seems appropriate in the field of radioecology. However, the manuscript's results and discussion are difficult to follow, especially when the author mixes up the results and discussion.

(1) Therefore, it is suggested that the author separate the results and discussion to make it easier and more comprehensible for readers to follow.

(2) The presentation of the results also seems insufficient. As most of the results are not directly presented in a table and mostly discussed in the text, I suggest the author add one table of descriptive statistics (n, mean +/- standard error, confidence intervals for mean, min-max) for each category (types of organs, year of collection, age, sex).

(3) In the discussion, I would like the author to add the point that Cs-137 tends to show high accumulation in muscle due to its high potassium K content. Cesium (Cs) and Potassium (K) have some similarities in their biodynamic patterns because they belong to the same group in the periodic table, Group 1 (alkali metals), and due to the fact that muscle tissue, such as skeletal muscles, contain significant amounts of potassium for metabolic functions. Similar to Sr-90 and calcium (Ca). This seems to correlate with the findings of the work where muscle (diaphragm and tongue) show high activities. Please cite this using appropriate references as well as ICRP Publication 110 Realistic reference phantoms.

Specific Comments:

Keywords: Keywords are crucial for increasing the visibility of the article and guiding readers. Consider including terms such as "Cesium 137-contaminated wild boar," "Cs-137 contaminated animal organ," "Cs-137 in Sus Scrofa," "wild boar radioactive bioindicator," and "North-western Italy Cesium-contamination."

Abstract: Limited values of results are given as most of the results are discussed. Add important values of the obtained results, e.g., mean, min-max based on different organ sites. This is important as the abstract needs to reflect the title, i.e., Cs in wild boar, but none of the Cs activity values are presented. Add a brief line about the main instrument used in this work. All approaches and methods are well described. Discussion and conclusion are sufficient.

Introduction: Revise the introduction to reflect the revised title and aim of the study.

Study Site: Revise the map in Fig. 1. Please title the countries in the map.

Line 177 - please elaborate on the process of homogenization. What matrix of the sample was obtained in this work? Is it in wet weight?

Line 229 - define ARPA, as it is missing in affiliation and not all readers recognize Agenzia Regionale per la Protezione dell'Ambiente.

Line 252-258 - metric SI prefix for (10^3) k or K?

Line 233 - superscript m^2

Line 245 - "Bq/k" does not appear to be a standard unit for specific activity, and it may be a typographical error.

Line 296 - p significant symbol. Sometimes you use small "p" and sometimes you use capital "P." Please be consistent.

Line 319 - sometimes "137Cs," sometimes "137^Cs," be consistent.

Fig. 2 missing unit on the y-axis.

Supplementary materials 1 & 2 - revise the title of each column. Typos and capital/small letter issues. The symbol for kilogram is "kg," not "Kg."

That's all.

Reviewers' comments:

Reviewer's Responses to Questions

**Comments to the Author**

1. If the authors have adequately addressed your comments raised in a previous round of review and you feel that this manuscript is now acceptable for publication, you may indicate that here to bypass the “Comments to the Author” section, enter your conflict of interest statement in the “Confidential to Editor” section, and submit your "Accept" recommendation.

Reviewer #3: All comments have been addressed

Reviewer #5: (No Response)

2. Is the manuscript technically sound, and do the data support the conclusions?

Reviewer #3: Yes

Reviewer #5: (No Response)

3. Has the statistical analysis been performed appropriately and rigorously? 

Reviewer #3: I Don't Know

Reviewer #5: (No Response)

4. Have the authors made all data underlying the findings in their manuscript fully available?

Reviewer #3: Yes

Reviewer #5: (No Response)

5. Is the manuscript presented in an intelligible fashion and written in standard English?

Reviewer #3: Yes

Reviewer #5: (No Response)

6. Review Comments to the Author

Reviewer #3: (No Response)

Reviewer #5: (No Response)

7. PLOS authors have the option to publish the peer review history of their article (what does this mean?). If published, this will include your full peer review and any attached files.

Reviewer #3: No

Reviewer #5: **Yes: **Tibor Kovacs

---

## [Author Response · Author response to Decision Letter 1]

12 Apr 2024

Dear Editors and Reviewers,

We would like to submit the manuscript entitled “137Cesium (137Cs) assessment in wild boars from northwestern Italy.” Pattono, D., Mannelli, A., Dalmasso A., Orusa, R., Faure Ragani, M., & Bottero M.T. for possible publication in Plos ONE.

We believe that the paper fits the aims and scope of the Journal. Indeed, it is an original article based on a survey of radionuclide contamination in two valley districts in Piemonte (Italy) caused by the Chernobyl and Fukushima nuclear power plant accidents. In particular, the study supplies the data missing on wild boar contamination, as needed by EU directives, and provides some insights into the distribution of radionuclide contamination across different organs in the body.

Briefly, the aim of the study was to provide data on 137Cs contamination in two valley districts in the Piedmont by sampling wild boars hunted for human consumption in two different years (2014 and 2016). The results indicate very low contamination levels in the two valleys studied compared with other valleys in the Piedmont. Moreover, the distribution of 137Cs between organs obtained from the same animal differed according to the age of the animal, with greater differences in younger animals. The diaphragm was the organ showing the highest levels of Cs137.

In conclusion, our analysis shows the lack of any significant risk of radionuclide contamination associated with the consumption of wild boar meat hunted in these valley districts. Moreover, the data indicate muscle, specifically the diaphragm, to be the most useful and convenient for sampling considering that this organ is already routinely collected by hunters for trichinellosis surveillance. Last, samples from younger animals may be more indicative for assessing the risk of meat consumption to human health associated with nuclear accidents, although further data will be needed to confirm this finding.

We would like to thank the Editor and the Reviewers for their comments, which have helped us to elevate the quality of our work.

For us, it is a great opportunity to improve the content, clarity and style of our work according to the comments provided by them. We have answered every concern point by point, and the file is enclosed in the submission page.

We uploaded the revised version of the manuscript on the submission page of the journal.

In accordance with the suggestions and comments of the Reviewers, the main changes we have made are as follows:

- Results and Discussion have been separated in order to improve comprehension and make easier for the reader to follow as requested by Reviewer 3;

- Tables of descriptive statistics has been added (Table n. 2 and n. 3) or modified (Table n. 1) as requested by reviewer 3;

- In the discussion we added a point stressing that 137Cesium belongs to the same periodic table’s group of potassium (K). This fact, as suggested by reviewer 3, can easily explain why muscle sites show high accumulation rates. For this reason, two more references have been cited in the text and added in the reference list. We thank the reviewer for this important suggestion;

- The keywords have been replaced with the suggested ones;

- Abstract: The abstract has been rewritten. We added more results and a new line about the instruments used for the cesium quantification as requested by Reviewer 3;

- Introduction: a paragraph has been added, as suggested by the reviewer 3, to improve the quality of the section and to be more linked to the aim of the work;

- The map has been revised and we added the name of the countries;

- The homogenization process was rewritten accordingly to the reviewer’s 3 suggestion;

- Here and in other parts of the paper we reported ARPA as Agenzia Regionale per la Protezione dell’Ambiente;

- We revised the metric prefix- We wrote in the corrected form the m^2;

- We wrote in the correct form “Bq/kg”;

- P symbol was corrected and unified throughout all the text

- 137Cesium was unified throughout all the text

- Fig. 2 unit was added on the y-axis

- In Supplementary Material 1 & 2 the titles have been corrected.

We confirm that neither the manuscript nor any parts of its content are currently under consideration or published in another journal. All authors have approved the manuscript and agree with its submission to Plos ONE. All figures and tables were produced by the authors. Last, all authors declare no conflicting interests.

In the hope that the manuscript fulfils the scientific standards of Plos ONE all the requests of the reviewers,

yours sincerely,

Daniele Pattono and coauthors

---

## [Editor Report · Decision Letter 2]

18 Apr 2024

137Cesium (137Cs) assessment in wild boars from northwestern Italy

PONE-D-23-24864R2

Dear Dr. Pattono,

We’re pleased to inform you that your manuscript has been judged scientifically suitable for publication and will be formally accepted for publication once it meets all outstanding technical requirements.

Kind regards,

Mohamad Syazwan Mohd Sanusi

Academic Editor

PLOS ONE
---

## [Editor Report · Acceptance letter]

25 Apr 2024

PONE-D-23-24864R2 

PLOS ONE

Dear Dr. pattono, 

I'm pleased to inform you that your manuscript has been deemed suitable for publication in PLOS ONE. Congratulations! Your manuscript is now being handed over to our production team.

Kind regards, 

on behalf of

Dr. Mohamad Syazwan Mohd Sanusi 

Academic Editor

PLOS ONE